# Characterization of tRNA-Derived Fragments in Lung Squamous Cell Carcinoma with Respect to Tobacco Smoke

**DOI:** 10.3390/ijms24065501

**Published:** 2023-03-13

**Authors:** Shruti Magesh, Pranava Gande, Rishabh Yalamarty, Daniel John, Jaideep Chakladar, Wei Tse Li, Weg M. Ongkeko

**Affiliations:** 1Department of Surgery, Division of Otolaryngology-Head and Neck Surgery, UC San Diego School of Medicine, San Diego, CA 92093, USA; 2Research Service, VA San Diego Healthcare System, San Diego, CA 92161, USA; 3School of Medicine, University of California, San Francisco, CA 94143, USA

**Keywords:** tRFs, LUSC, biomarker

## Abstract

Lung squamous cell carcinoma (LUSC) is a highly heterogeneous cancer that is influenced by etiological agents such as tobacco smoke. Accordingly, transfer RNA-derived fragments (tRFs) are implicated in both cancer onset and development and demonstrate the potential to act as targets for cancer treatments and therapies. Therefore, we aimed to characterize tRF expression with respect to LUSC pathogenesis and clinical outcomes. Specifically, we analyzed the effect of tobacco smoke on tRF expression. In order to do so, we extracted tRF read counts from MINTbase v2.0 for 425 primary tumor samples and 36 adjacent normal samples. We analyzed the data in three primary cohorts: (1) all primary tumor samples (425 samples), (2) smoking-induced LUSC primary tumor samples (134 samples), and (3) non-smoking-induced LUSC primary tumor samples (18 samples). Differential expression analysis was performed to examine tRF expression in each of the three cohorts. tRF expression was correlated to clinical variables and patient survival outcomes. We identified unique tRFs in primary tumor samples, smoking-induced LUSC primary tumor samples, and non-smoking-induced LUSC primary tumor samples. In addition, many of these tRFs demonstrated correlations to worse patient survival outcomes. Notably, tRFs in the smoking-induced LUSC and non-smoking-induced LUSC primary tumor cohorts were significantly correlated to clinical variables pertaining to cancer stage and treatment efficacy. We hope that our results will be used to better inform future LUSC diagnostic and therapeutic modalities.

## 1. Introduction

Lung cancer is the second most common cancer in the world and accounts for over 1.7 million deaths each year [1]. The two main subtypes of lung cancer are non-small cell lung cancer (NSCLC) and small cell lung cancer (SCLC). NSCLC accounts for over 85% of all lung cancer cases and is divided into two primary classifications: lung adenocarcinoma (LUAD) and lung squamous cell carcinoma (LUSC) [2,3,4]. LUSC is considered a very aggressive cancer and is often only detected following metastasis. While the average five-year survival rate for LUSC is 24%, survival rates can be as high as 99% when the cancer is detected early [5,6]. Accordingly, LUSC has been found to arise in the proximal airways and bronchioles and is closely associated with chronic inflammation and risk factors such as tobacco smoke and alcohol consumption [7,8]. Thus, due to the low survival rates and heterogeneous nature of LUSC, it is critical to examine the role of etiological agents and molecular factors on LUSC pathogenesis and metastasis to develop more accurate therapeutic and diagnostic modalities to improve cancer detection and prevention.

Transfer RNAs (tRNAs) are small noncoding RNA molecules that play critical roles in protein synthesis and interact very closely with messenger RNAs (mRNAs) and amino acids [9]. Specifically, tRNAs attach to a base on the mRNA chain and ensure that the corresponding amino acid for that particular base is added to the growing protein chain [9]. During maturation, tRNAs undergo various cleavages and enzymatic splicing events that give rise to new classifications of tRNAs, which include: tRNA-derived small RNAs (tsRNAs), tRNA-derived stress-induced RNAs (tiRNAs), and tRNA-derived fragments (tRFs) [10]. In particular, tRFs are small microRNA (miRNA) sized molecules that play important regulatory and functional roles in various diseases [11]. Previous studies have suggested that tRFs may be implicated in cancer pathogenesis and proliferation and may act as biomarkers for cancer prognosis [12,13]. For instance, a study examining tRF expression across various cancers found that tRF-20-S998LO9D was differentially expressed in several cancers and was associated with reduced cell proliferation in LUSC [13]. Another study examining tRF expression on a pan-cancer scale observed that tRF-28-RS9NS334L2DB was associated with the clinical tumor stage in LUSC [14]. Similarly, tRF-21 levels were correlated to shorter patient survival rates [15]. In addition, 3′ half tRFs were more likely to be associated with LUSC and LUAD when compared to the expression of other tRFs [14]. As such, these studies suggest that tRFs may play critical roles in carcinogenesis. However, despite the potential of tRFs to act as biomarkers for cancer diagnosis and treatment, relatively few studies comprehensively analyze and identify a panel of tRFs representative of LUSC prognosis and clinical outcomes. 

LUSC development is highly correlated to tobacco smoke, with over 80% of LUSC patients being considered former or reformed smokers [16]. As such, tobacco smoke acts as the primary etiological agent of LUSC. A wide range of evidence has proved that tobacco smoke contains over 3500 chemicals and 60 carcinogens, which include but are not limited to volatile organic hydrocarbons, polycyclic aromatic hydrocarbons, and *N*-Nitrosamines [17]. Moreover, some of the chemicals in tobacco smoke are considered to be reactive oxygen species (ROS), which can promote inflammation and mediate processes such as cell proliferation, cell cycle progression, and metastasis [17,18]. Accordingly, data have demonstrated that the byproducts of the chemicals and compounds in tobacco smoke are found in the urine of smokers at higher levels than that of non-smokers. When taken in, these compounds have the potential to form DNA adducts, which can cause somatic mutations [17]. If mutations occur in tumor suppressor genes or oncogenes such as *p53* or *RAS*, cell growth control mechanisms may be lost, thus resulting in cancer onset and development [19]. Accordingly, smoking is correlated to increased mutations and copy number variants in LUSC [20]. For instance, smoking was associated with *TP53* mutations in LUSC patients [16]. Similarly, *KRAS*, *p53*, and *BRAF* mutations, as well as mutations in the RAS/Rtk pathway, were found to be enriched in smokers when compared to non-smokers in NSCLC [21]. Previous research has also demonstrated that smoking is associated with alterations in the expression of long noncoding RNAs (lncRNA) with respect to lung cancer prognosis and survival rates [22]. Moreover, the expression of immune cells, including CD4 T cells and NK cells, was observed to be differentially regulated in smoking-induced LUSC patients when compared to non-smoking-induced LUSC patients, which may play a role in promoting tumor progression and altering interactions within the tumor microenvironment [20]. Furthermore, we have conducted studies examining the implications of enhancer RNAs (eRNA) in LUSC pathogenesis with respect to the following etiological agents: tobacco smoke and electronic cigarette (e-cigarette) smoke [23]. Thus, the differential regulation of these molecular factors in the presence of tobacco smoke may act to directly contribute to LUSC onset and development. Accordingly, tRFs have demonstrated significant implications in cancer development and clinical outcomes. However, very few studies have characterized the mechanisms by which tobacco smoke may regulate the expression of tRFs with respect to LUSC. 

In this study, we aim to comprehensively characterize tRF expression in LUSC. We also aim to investigate the effect of tobacco smoke on tRF expression. To do so, we will analyze correlations between tRF expression, patient survival outcomes, and clinical variables to determine how these tRFs may contribute to LUSC pathogenesis and proliferation. In order to do so, we examined 425 LUSC samples from MINTbase v2.0 [24]. Specifically, we analyzed data from 134 smoking and 18 non-smoking LUSC patients. We utilized differential expression analysis to identify significantly differentially expressed tRFs associated with both smoking-induced LUSC primary tumor samples and non-smoking-induced LUSC primary tumor samples. Next, we performed survival analysis and clinical variable correlations to determine whether these significantly dysregulated tRFs were associated with patient survival outcomes and clinical factors such as cancer pathologic stage. To the best of our knowledge, we are the first to comprehensively characterize tRF expression in LUSC with respect to tobacco smoke. We hope that our findings can better inform future cancer diagnostic and therapeutic modalities.

## 2. Results

### 2.1. Differential Expression and Survival Correlations of tRFs in LUSC

In order to analyze the expression of tRFs in LUSC, we performed differential expression analysis on primary tumor samples and adjacent normal samples (*p*-value < 0.05 and |log fold change (FC)| > 1). We identified 10 significantly differentially expressed tRFs when comparing tumor samples to normal samples. AsnATT 3′-tRF, AsnGTT 5′-tRF, ArgTCT 5′-tRF, ArgTCG 5′-tRF, and AlaAGC 5′-tRF were upregulated in tumor samples, while LeuCAA 3′-tRF, ThrAGT 3′-tRF, GlyTCC 3′-tRF, AlaAGC i-tRF, and ArgCCT 5′-tRF were downregulated in tumor samples (Table 1, Figure 1). 

We next analyzed the correlations between tRF expression and patient survival outcomes using Kaplan–Meier survival analysis and plotted the results using Cox proportional hazards regression, with a *p*-value of 0.05 indicating a significant result. We found that 10 tRFs were significantly correlated to patient survival outcomes. Specifically, the upregulation of AlaCGC 5′-tRF was associated with worse patient survival outcomes, while the downregulation of TyrGTA i-tRF, GlyGCC i-tRF, LeuAAG 3′-tRF, LysTTT 5′-half, CysGCA 5′-tRF, TrpTCA 3′-tRF, ArgTCG 5′-tRF, and LeuTAA i′-tRF was associated with worse patient survival outcomes (Figure 2). Notably, ArgTCG 5′-tRF was both significantly differentially expressed in tumor samples and implicated in worse patient survival outcomes, suggesting its potential to act as a biomarker of worse clinical outcomes in LUSC (Figure 2). However, further studies are required to fully elucidate this finding.

### 2.2. Differential Expression and Survival Correlations of tRFs in Smoking-Induced LUSC Primary Tumor Samples and Non-Smoking-Induced LUSC Primary Tumor Samples

In order to analyze the effect of tobacco smoke on tRF expression, we examined 134 smoking-induced LUSC primary tumor samples and 18 non-smoking-induced LUSC primary tumor samples. We conducted analyses on these two cohorts separately, with adjacent normal samples as the control group. Using differential expression analysis (*p*-value < 0.05 and |log (FC)| > 1), we identified 12 tRFs to be differentially expressed in smoking-induced LUSC primary tumor samples. AsnATT 3′-tRF, SerACT 3′-tRF, AsnGTT 5′-tRF, CysACA 5′-tRF, GlyGCC 5′-tRF, and AsnGTT 3′-tRF were upregulated in tumor samples, while ThrAGT 3′-tRF, LeuCAA 3′-tRF, LeuAAG i-tRF, GlyTCC 3′-tRF, IleGAT i-tRF, and AlaAGC i-tRF were downregulated in tumor samples. We identified nine tRFs to be differentially expressed in non-smoking-induced LUSC primary tumor samples, with three upregulated tRFs and six downregulated tRFs in non-smoking-induced LUSC tumor samples. Interestingly, GlyGCC 5′-tRF and GlyTCC 3′-tRF were found to be differentially expressed in both smoking-induced LUSC primary tumor samples and non-smoking-induced LUSC primary tumor samples, indicating their potential to influence LUSC onset and development independent of etiological agents (Figure 3). 

We next assessed the correlations between tRF expression and patient survival in the smoking-induced LUSC and non-smoking-induced LUSC primary tumor sample cohorts. We identified 24 tRFs that were associated with worse patient survival outcomes in the smoking-induced LUSC primary tumor samples. Specifically, the upregulation of 15 tRFs and the downregulation of 9 tRFs were correlated to worse patient survival outcomes. In the non-smoking-induced LUSC primary tumor samples, we identified six upregulated tRFs and one downregulated tRF associated with worse patient survival outcomes. Through this analysis, we were able to identify unique tRFs that were associated with worse patient prognosis for non-smoking-induced LUSC primary tumor samples and smoking-induced LUSC primary tumor samples. The identified tRFs demonstrate the potential to act as biomarkers for patient prognosis in each of these respective cohorts. However, further studies are required to elucidate the clinical applications of these molecular markers (Figure 3).

### 2.3. tRF Correlations to Clinical Variables in Smoking-Induced LUSC and Non-Smoking-Induced LUSC Primary Tumor Samples

Previous studies have suggested that tRFs are implicated in cancer pathogenesis and proliferation [13]. As such, we aimed to assess whether tRFs were associated with various clinical features in LUSC. Specifically, we examined correlations for the following clinical variables: residual tumor, cancer pathologic stage, cancer pathologic TNM stage, cancer neoplasm status, new tumor event after treatment, and primary therapy outcome. We used the Kruskal–Wallis test in the edgeR library to perform this analysis, with a *p*-value < 0.05 indicating a significant result. Correlations were conducted separately for the following three cohorts: (1) LUSC primary tumor samples, (2) smoking-induced LUSC primary tumor samples, and (3) non-smoking-induced LUSC primary tumor samples. We did not observe any significant correlations for the LUSC primary tumor sample cohort. 

Our results demonstrated that 12 tRFs were associated with cancer pathologic stage in smoking-induced LUSC primary tumor samples, while only 6 tRFs were associated with cancer pathologic stage in non-smoking-induced LUSC primary tumor samples. In general, we observed increased tRF expression in cancer stages 1a, 2a, and 2b in smoking-induced LUSC primary tumor samples, while increased tRF expression was associated with cancer stage 2a and 2b in non-smoking-induced LUSC primary tumor samples. However, it is important to note that these are the general trends we observed and that specific tRFs were associated with cancer pathologic stages in different ways.

We observed similar amounts of tRFs associated with cancer pathologic stage m across both cohorts, with five tRFs in smoking-induced LUSC and six tRFs in non-smoking-induced LUSC. Overall, we observed that there was increased expression of tRFs in stage m0 when compared to stage m1 in both non-smoking and smoking-induced LUSC primary tumor samples. Interestingly, we observed the greatest number of correlations to cancer pathologic stage n in non-smoking-induced LUSC primary tumor samples, with eight tRFs, while smoking-induced LUSC primary tumor samples only demonstrated three tRF correlations to this particular variable. There was increased expression of tRFs in stages n0, n1, and n2 in smoking-induced LUSC primary tumor samples, while there was increased expression of tRFs in stages n1 and n2 in non-smoking-induced LUSC primary tumor samples. Conversely, we observed that cancer pathologic stage t had the most correlations to smoking-induced LUSC primary tumor samples with 11 tRFs, while non-smoking-induced LUSC primary tumor samples only demonstrated 5 tRF correlations to this variable (Figure 4). We observed varying levels of tRF expression across cancer pathologic stage t, with certain tRFs being highly expressed in specific t stages. Overall, we observed increased tRF expression in stages t1, t2, and t4 in smoking-induced LUSC primary tumor samples, while we primarily observed elevated tRF expression in stage t1a and t2b in non-smoking-induced LUSC primary tumor samples.

We also observed that the following clinical variables had significant tRF correlations: cancer neoplasm status, new tumor event following treatment, and residual tumor. It is important to note that each patient received different treatments in our dataset. The overall percentages of cancer patients who received each initial drug treatment are as follows: 0.24% received adriamycin, 0.24% received alimta, 6.59% received carboplatin, 10.12% received cisplatin, 0.24% received docetaxel, 0.24% received eloxatin, 0.24% received erlotinib, 0.47% received etoposide, 1.65% received gemcitabine, 0.94% received gemzar, 0.24% received mithramycin, 0.47% received navelbine, 0.94% received paclitaxel, 0.24% received paraplatin, 1.65% received taxol, 0.71% received taxotere, 0.47% received vepesid, 1.41% received vinorelbine, and the remaining patients received other clinical trial drugs and/or did not have treatment type information. A total of 25.18% of all cancer patients received chemotherapy as well. Data regarding surgical treatments were not clear.

Both new tumor events following treatment, with 18 tRF correlations, and patient residual tumor, with 8 tRF correlations, demonstrated more correlations in smoking-induced LUSC primary tumor samples when compared to non-smoking-induced LUSC primary tumor samples. In contrast, cancer neoplasm status had more tRF correlations in the non-smoking-induced LUSC primary tumor samples (Figure 4). 

More specifically, the increased expression of AlaCGC 5′-tRF and AsnGTT 3′-tRF was associated with new tumor events following treatment, suggesting that these tRFs may play a role in tumor recurrence. In contrast, the increased expression of GlnCTG 5′-tRF, LeuAAG 5′-tRF, and LeuCAA 5′-tRF was associated with no new tumor events following treatment, indicating that they may act as biomarkers for favorable outcomes following treatment. We also observed that IleTAT 5′-tRF was associated with patient residual tumor stage R1. R1 residual tumors are classified as a microscopic residual disease. Accordingly, past research has found that patients with R1 residual tumors have a significantly decreased survival rate of 14% in NSCLC [25]. Therefore, we predict that IleTAT 5′-tRF may play regulatory roles in residual tumor occurrence (Figure 4). 

Notably, we observed significant clinical variable correlations in five of the tRFs that we previously identified to be differentially expressed in smoking-induced LUSC primary tumor samples, suggesting that these tRFs may play critical roles in cancer malignancy and may be useful as biomarkers of cancer stage and metastasis. In particular, GlyGCC 5′-tRF was associated with a lack of new tumor events following treatment, demonstrating its potential to act as a biomarker of favorable clinical outcomes in smoking-induced LUSC. 

## 3. Discussion

In this study, we identified a panel of tRFs with implications in LUSC pathogenesis and patient clinical outcomes. We additionally correlated tobacco smoking status to tRF expression. Previous studies have shown that tRFs are associated with tumor stage and patient survival rates in LUSC [12,13,14,15]. Thus, it appears that tRFs may play significant roles in cancer development. Accordingly, LUSC is a very heterogeneous cancer that is highly influenced by the presence of etiological agents such as tobacco smoke [16]. Studies have shown that tobacco smoke increases somatic mutations in select genes, which may subsequently promote carcinogenesis [17]. In particular, past research has demonstrated that several molecular factors, such as eRNA, lncRNAs, and immune cells, may act as mediators of tobacco smoke-induced LUSC development [22,23]. However, despite the implications of these molecular factors on cancer malignancy, relatively few studies analyze tRF expression in LUSC with respect to etiological agents such as tobacco smoke. To the best of our knowledge, we are the first to comprehensively profile tRF expression in LUSC primary tumor samples, smoking-induced LUSC primary tumor samples, and non-smoking-induced LUSC primary tumor samples. 

In order to do so, we extracted tRF counts from MINTbase v2.0 for 425 primary tumor samples and 36 adjacent normal samples [24]. We specifically analyzed the data in three primary cohorts, with adjacent normal samples as the control group: (1) LUSC primary tumor samples (425 samples), (2) smoking-induced LUSC primary tumor samples (134 samples), and (3) non-smoking-induced LUSC primary tumor samples (18 samples). 

Using differential expression analysis, we identified 10 significantly differentially expressed tRFs in the LUSC primary tumor samples when compared to normal samples. Of these 10 differentially expressed tRFs, 5 tRFs were upregulated in tumor samples, while 5 tRFs were downregulated in tumor samples. Next, we examined tRF correlations to patient survival outcomes. While we identified 10 tRFs to be associated with worse overall patient prognosis, we found that ArgTCG 5′-tRF was both differentially expressed in tumor samples and associated with worse patient survival outcomes. Accordingly, previous pan-cancer studies on tRF expression have found that 5′-ArgTCG-3-1-L19 tRF has been dysregulated in several cancers, including head and neck squamous cell carcinoma (HNSCC), kidney renal cell carcinoma (KIRC), liver hepatocellular carcinoma (LIHC), and NSCLC [26]. Furthermore, 5′ tRFs have been found to be highly associated with carcinogenesis. For instance, a previous study has implicated 5′-IleAAT-8-1-L20 in the regulation of cancer development and metastasis in lung cancer [26]. Similarly, other studies observed that 5′-tRF-LysCTT was associated with advanced tumor phenotype and worse response to treatment in bladder cancer, while tRF-Leu-CAG promoted cell proliferation through the regulation of AURKA in NSCLC [27,28]. Thus, our findings appear to be consistent with past research, and as such, we predict that ArgTCG 5′-tRF may act as a potential biomarker for unfavorable clinical outcomes in LUSC. 

Next, we analyzed the effect of tobacco smoke on tRF expression in relation to LUSC. To do so, we compared tRF expression in smoking-induced LUSC and non-smoking-induced LUSC primary tumor samples. We identified 12 differentially expressed tRFs in smoking-induced LUSC primary tumor samples, with 6 upregulated and 6 downregulated tRFs in tumor samples. Conversely, we identified nine differentially expressed tRFs in non-smoking-induced LUSC primary tumor samples, with three upregulated tRFs and six downregulated tRFs in tumor samples. Moreover, using survival analysis, we found that 24 tRFs were associated with worse patient survival outcomes in smoking-induced LUSC primary tumor samples, while only 7 tRFs were associated with worse patient survival outcomes in non-smoking-induced LUSC primary tumor samples. Thus, our results suggest that a greater number of tRFs are associated with worse prognosis in smoking-induced LUSC primary tumor samples when compared to non-smoking-induced LUSC primary tumor samples, demonstrating that these tRFs may play critical regulatory roles in mediating tobacco smoke-mediated carcinogenesis. Very few studies examine the modulation of tRF expression by etiological agents. However, past research has demonstrated that tobacco smoke and alcohol regulate the expression of noncoding RNAs, such as lncRNAs and miRNAs, through apoptotic and inflammatory pathways [29]. These pathways are subsequently implicated in a wide variety of diseases, including cancer. Our results provide novel insights regarding the modulation of tRFs by tobacco smoke and their implications in LUSC prognosis. 

In order to further examine the roles of tRFs in cancer pathogenesis and metastasis, we performed clinical variable correlations for tRFs in each of our three previously defined cohorts. We did not observe significant clinical variable correlations in the primary tumor cohort. 

In smoking-induced LUSC primary tumor samples, we found that AlaCGC 5′-tRF and AsnGTT 3′-tRF were associated with new tumor events following treatment, while GlnCTG 5′-tRF, LeuAAG 5′-tRF, and LeuCAA 5′-tRF were associated with no new tumor events following treatment. Thus, it appears that these particular tRFs may provide insight into treatment efficacy. Previous studies have found that tRFs mediate several signaling pathways that may affect response to chemotherapy treatments in lung cancer. For example, tRFs can bind to mRNA in place of eukaryotic translation initiation factor 4G (*eIF4G*), thus inhibiting protein translation and expression [30,31]. The downregulation of *eIF4G* is also associated with the increased sensitivity of cancer drugs, such as cisplatin, in NSCLC [30,31]. tRFs also promote the formation of stress granules, which function to prevent mRNA damage during stressful conditions. However, these stress granules may also interfere with the apoptotic effects of established cancer treatments and are therefore involved in drug resistance [30,31]. Accordingly, drug resistance is closely associated with the formation of new tumors following treatment [32]. As such, we predict that AlaCGC 5′-tRF and AsnGTT 3′-tRF may act through these specific mechanisms to promote drug resistance and new tumor formation following treatment. However, further in vitro studies are required to fully elucidate these findings. 

In addition to analyzing clinical variables for treatment response, we examined clinical variables pertaining to tumor progression and metastasis. We found that 12 tRFs were associated with cancer pathologic stage in smoking-induced LUSC primary tumor samples, demonstrating the potential role of these tRFs in mediating tumor progression. Interestingly, we observed similar amounts of tRF correlations with cancer pathologic stage m in both smoking-induced LUSC and non-smoking-induced LUSC primary tumor samples. However, we observed that non-smoking-induced LUSC primary tumor samples had the greatest amount of clinical variable correlations to cancer pathologic stage n, while smoking-induced LUSC primary tumor samples had the greatest amount of clinical variable correlations to cancer pathologic stage t. Therefore, as more tRFs were associated with cancer pathologic stage n in non-smoking-induced LUSC primary tumor samples, it appears that tRFs may play larger roles in promoting cancer progression in non-smoking-induced LUSC primary tumor samples when compared to smoking-induced LUSC primary tumor samples. However, more extensive studies are required to fully understand this particular finding. 

In all, we identified tRFs unique to LUSC primary tumor samples, smoking-induced LUSC primary tumor samples, and non-smoking-induced LUSC primary tumor samples. We also examined the implications of these tRFs on clinicopathological variables and patient survival outcomes. Our results suggest that tRF expression varies greatly across smoking-induced LUSC and non-smoking-induced LUSC primary tumor samples, indicating that tobacco smoke may modulate tRF expression to promote cancer onset and development. We hope that our results provide a more specific panel of molecular biomarkers for novel LUSC diagnostic and therapeutic modalities with respect to etiological agents. However, it is important to note that this study is computational in nature, and as such, in vitro experiments are required to confirm the findings presented in this study. 

## 4. Materials and Methods

### 4.1. Data Acquisition

In order to assess tRF expression in LUSC, we extracted tRF read counts from MINTbase v2.0 for 425 primary tumor samples and 36 adjacent normal samples. The tissue keyword “LUSC” was used to search in MINTbase v2.0 on the tRF expression profile page. A zip file containing the tRF expression for each patient was downloaded from MINTbase v2.0 (https://cm.jefferson.edu/tcga-mintmap-profiles/; accessed on 14 December 2022). We analyzed the data in three primary cohorts: (1) LUSC primary tumor samples (425 samples), (2) smoking-induced LUSC primary tumor samples (134 samples), and (3) non-smoking-induced LUSC primary tumor samples (18 samples) [24]. Primary tumor samples were defined as the original tumor that developed in the patient and did not include metastasized cells. The primary tumor sample cohort refers to all primary tumor samples without stratification by smoking status. Accordingly, the smoking-induced LUSC primary tumor samples and the non-smoking-induced LUSC primary tumor samples include the same primary tumor samples but also stratified the samples by smoking status, allowing for a more specific subset of samples. Adjacent normal LUSC samples served as the control group for each of these three cohorts. MINTbase v2.0 had a low amount of control samples available for analysis. Thus, our study had a limited amount of control samples. The smoking-induced LUSC primary tumor cohort only contained current smokers to limit the effect of confounding variables, while the non-smoking-induced LUSC primary tumor cohort only included lifelong non-smokers. Former smokers were excluded from each of these respective cohorts, as former smokers demonstrate reduced risks of cancer incidence when compared to current smokers [33]. Patient clinical variable data were extracted from the Broad Institute GDAC Firebrowse Database [24]. 

### 4.2. Differential Expression Analysis

Differential expression analysis was conducted in the edgeR library to compare tRF expression across the following cohorts: (1) primary tumor samples, (2) smoking-induced LUSC primary tumor samples, and (3) non-smoking-induced LUSC primary tumor samples. Adjacent normal samples were used as the control group for this analysis. Significantly differentially expressed tRFs were identified and filtered based on log fold change and *p*-value (|log fold change (FC)| > 1 and *p*-value < 0.05).

### 4.3. tRF Correlations to Clinical Variables and Patient Survival

tRF expression was further analyzed for correlations to patient clinical variables in the previously defined cohorts using the Kruskal–Wallis test in the edgeR library. Specifically, we examined correlations to the following clinical variables from The Cancer Genome Atlas (TCGA): residual tumor, cancer pathologic stage, cancer pathologic TNM stage, cancer neoplasm status, new tumor event after treatment, and primary therapy outcome. Significant correlations were filtered by *p*-value < 0.05. Clinical variables that contained three or more groups were adjusted so that each plot only showed correlations between two sets of variables at a time so that the *p*-value would be an accurate measure of statistical significance. The groups that were adjusted for containing three or more groups include residual tumor and cancer pathologic TNM stage.

tRF expression was then correlated to patient survival outcomes using survival analysis in the edgeR library, and the results were plotted using the Cox proportional hazards regression. tRF read counts were used as the primary binary variable, with expression being described as above or below the median expression value. The following variables were used as measures of patient survival: patient time to death and patient status. These clinical variables were extracted from the Broad Institute GDAC Firebrowse Database [24]. Significant correlations were filtered by *p*-value < 0.05.

## Figures and Tables

**Figure 1 ijms-24-05501-f001:**
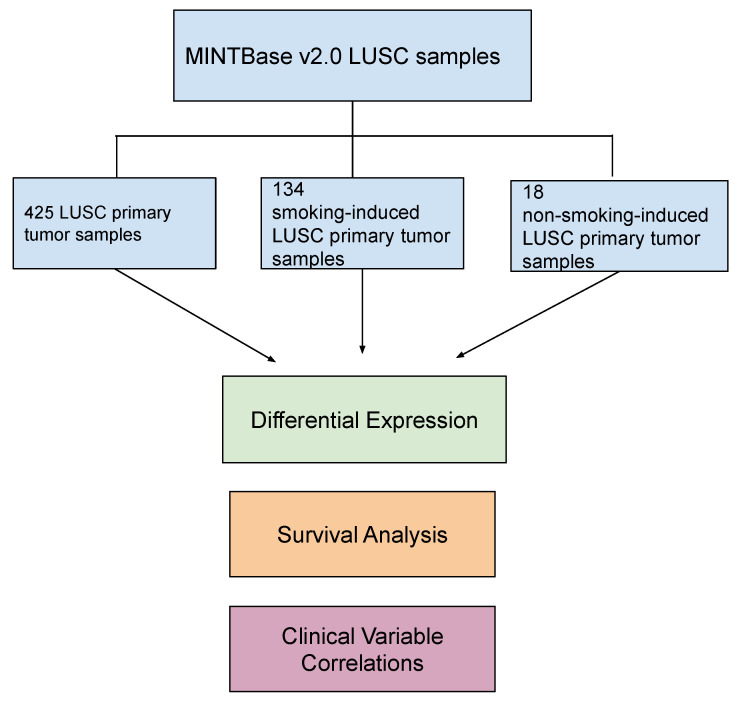
Schematic of project workflow. Transfer RNA-derived fragment (tRF) counts were extracted from MINTbase v2.0 (https://cm.jefferson.edu/tcga-mintmap-profiles/; accessed on 14 December 2022) for 425 lung squamous cell carcinoma (LUSC) primary tumor samples and 36 adjacent normal samples. We analyzed the data in three primary cohorts: (1) primary tumor samples (425 samples), (2) smoking-induced LUSC primary tumor samples (134 samples), and (3) non-smoking-induced LUSC primary tumor samples (18 samples). Adjacent normal LUSC samples were used as the control group. Differential expression analysis was performed to identify significantly differentially expressed tRFs in each of these cohorts (|log fold change (FC)| > 1 and *p*-value < 0.05). tRF correlations to patient survival outcomes and clinical variables were also performed to analyze the role of these tRFs in cancer onset and development (*p*-value < 0.05).

**Figure 2 ijms-24-05501-f002:**
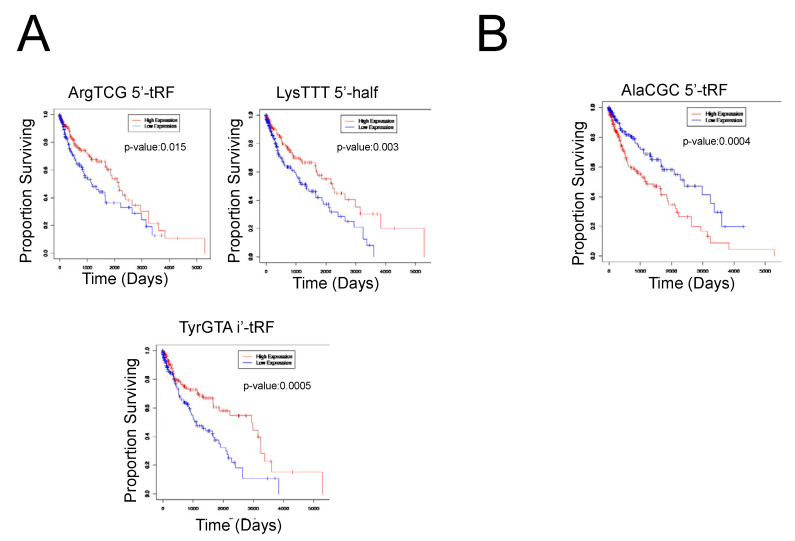
Clinical significance of tRFs in LUSC primary tumor samples. (**A**) Select survival plots demonstrating the decreased expression of the respective tRF with worse patient survival outcomes. The red curve represents high expression of the tRF, while the blue curve represents low expression of the tRF. When the high expression curve is above the low expression curve in the graph, it demonstrates that the decreased expression of the tRF is correlated to worse patient survival outcomes. (**B**) Select survival plot demonstrating the increased expression of the respective tRF with worse patient survival outcomes. The red curve represents high expression of the tRF, while the blue curve represents low expression of the tRF. When the high expression curve is below the low expression curve in the graph, it demonstrates that the increased expression of the tRF is correlated to worse patient survival outcomes.

**Figure 3 ijms-24-05501-f003:**
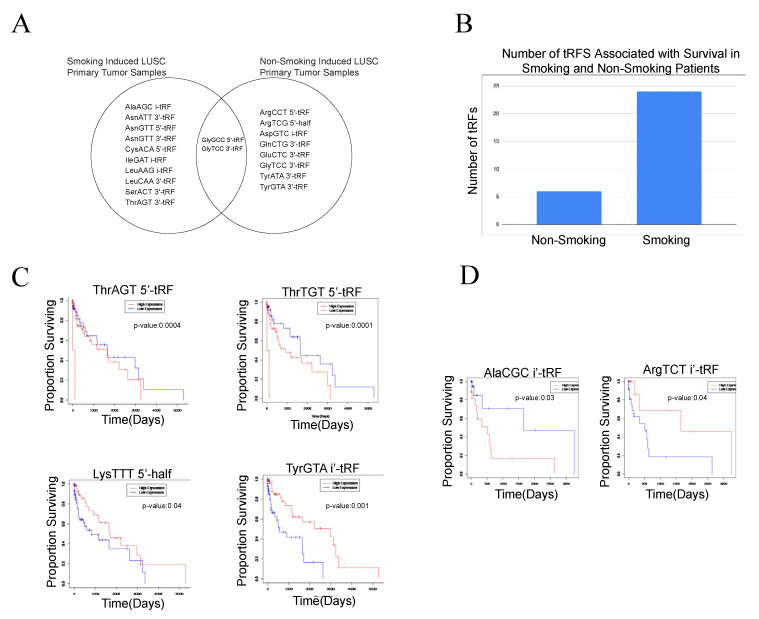
Differentially expressed tRFs and correlations to clinical variables in the smoking-induced LUSC primary tumor samples and non-smoking-induced LUSC primary tumor samples. Differential expression analysis and survival analysis were utilized to identify tRFs unique to smoking-induced LUSC primary tumor samples and non-smoking-induced LUSC primary tumor samples. Analyses were conducted separately for each of these cohorts. (**A**) Venn diagram comparing significantly differentially expressed tRFs in the smoking-induced LUSC primary tumor samples and non-smoking-induced LUSC primary tumor samples (*p*-value < 0.05). GlyGCC 5′-tRF and GlyTCC 3′-tRF were found to be differentially expressed in both smoking-induced LUSC and non-smoking-induced LUSC primary tumor samples. (**B**) Bar graph displaying the number of tRFs associated with survival in both smoking-induced LUSC and non-smoking-induced LUSC primary tumor samples. Specifically, a much larger number of tRFs were associated with smoking-induced LUSC primary tumor samples in comparison to non-smoking-induced LUSC primary tumor samples, demonstrating their potential to influence clinical outcomes. (**C**) Select survival plots demonstrating tRF correlations to patient survival in smoking-induced LUSC primary tumor samples. The red curve represents high expression of the tRF, while the blue curve represents low expression of the tRF. When the high expression curve is above the low expression curve in the graph, it demonstrates that the decreased expression of the tRF is correlated to worse patient survival outcomes. (**D**) Select survival plots demonstrating tRF correlations to patient survival in non-smoking-induced LUSC primary tumor samples.

**Figure 4 ijms-24-05501-f004:**
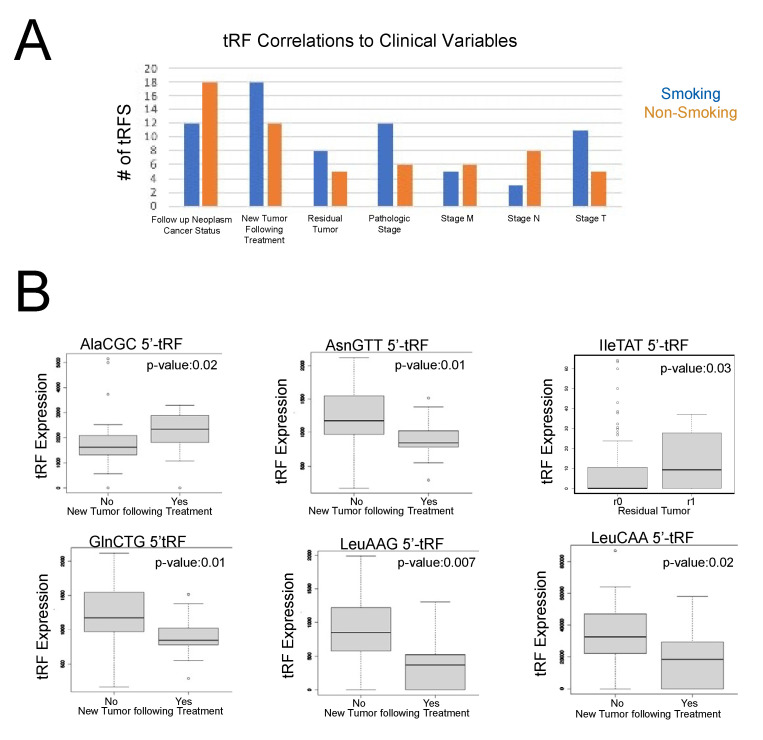
tRF correlations to clinical variables in smoking-induced LUSC and non-smoking-induced LUSC primary tumor samples. tRF correlations to clinical variables were performed using the Kruskal–Wallis test in the edgeR library (*p*-value < 0.05). Specifically, the following clinical variables were utilized for the purposes of this study: residual tumor, cancer pathologic stage, cancer pathologic TNM stage, cancer neoplasm status, new tumor event after treatment, and primary therapy outcome. (**A**) Bar graph representing the total number of tRFs associated with the seven clinical variables that displayed significant correlations to tRF expression. Orange represents non-smoking-induced LUSC primary tumor samples, while blue represents smoking-induced LUSC primary tumor samples. (**B**) Select box plots representing tRF correlations to clinical variables (*p*-value < 0.05). The x-axis represents a clinical variable, with specific subgroups for the variable, as noted in each graph. The y-axis represents tRF expression. Each box displays the maximum, minimum, median, lower quartile, and upper quartile of tRF expression for each condition. Specifically, by comparing the median tRF expression value between groups, we can draw conclusions regarding the role of tRFs in carcinogenesis and treatment response.

**Table 1 ijms-24-05501-t001:** tRFs Associated with LUSC Primary Tumor Samples.

Upregulated tRFs	Downregulated tRFs
AsnATT 3′-tRF	LeuCAA 3′-tRF
AsnGTT 5′-tRF	ThrAGT 3′-tRF
ArgTCT 5′-tRF	GlyTCC 3′-tRF
ArgTCG 5′-tRF	AlaAGC i-tRF
AlaAGC 5′-tRF	ArgCCT 5′-tRF

## Data Availability

Publicly available datasets were analyzed in this study. This data can be found here: https://cm.jefferson.edu/tcga-mintmap-profiles/ (accessed on 14 December 2022).

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
