# Peer review of "Characterization of tRNA-Derived Fragments in Lung Squamous Cell Carcinoma with Respect to Tobacco Smoke"

_ijms, 2023, doi:10.3390/ijms24065501_

Round 1

Reviewer 1 Report

This paper describes in silico analysis of tRF expression data and their correlation with clinicopatholoogical data in lung squamous cell carcinoma (LUSC) samples. The aim was to characterize tRF expression, with respect to LUSC pathogenesis and clinical outcomes and to identify tRFs that may have potential as prognostic/predictive biomarkers. The data analyzed were obtained from the publicly available MINTbase and Firebrowse Database. The results point to several tRF that might be associated with cancer development/progression and response to therapy.

The paper covers a new field of research but it falls short in several ways. Here are my comments:

Introduction:

-  page 2: In the sentence related to ref. 21, the authors name several genes ( KRAS, RAS, p53, and BRAF) that were found to harbor mutations in smokers compared to non-smokers. What is the difference between KRAS and RAS?

Materials and Methods

- In Materials and Methods section the authors should explain the search criteria and keywords used for the search of MINTBase.

- What are  "primary tumor samples" in the context of smoking? The meaning of "primary"? Aren't the smoking induced LUSC samples and  non-smoking induced LUSC samples also primary tumors?

- The group of adjacent normal samples (controls) consists of only 36 samples, it is rather small, particularly compared to the group of primary tumor samples- 425. More controls should be added to the study, or an explanation for a low number of control samples. 

Results:

- The resolution of 2B, 2C, 3C, and 4B should be improved

-  The name of the section "2.3. tRF Correlations to Clinical Variables in Smoking Induced LUSC" should be corrected since the authors present the results for the non-smoking induced LUSC samples as well

- figure 4B- What are the 2 or 3 groups that were compared for the expression levels of specific tRFs? The designations are unclear. Also, since there are three groups that are compared (one p value)  what is the significance of the differences between pairs of analyzed groups?

- the meaning of "tRFs associated with cancer pathologic stage, Stage M, N, T.. across cohorts) should be better explained... How are they associated- increased or decreased expression in advanced patologic stage/ M1, N1, higher T?

References:

- All references should be cited according to the citation style, for example:

 ref 3,5,6,9, 15,18, 21, 32 are not complete... they lack journal/book name, issue#, pages...

Reviewer 2 Report

The author analyzed correlations between tRF expression, patient survival outcomes, and clinical variables to determine how these tRFs may contribute to LUSC pathogenesis and proliferation. Specifically, they utilized differential expression analysis to identify significantly differentially expressed tRFs associated with both smoking-induced LUSC and non-smoking-induced LUSC from MINTbase v2.0. The result showed that tRFs in the smoking-induced LUSC and non-smoking-induced LUSC cohorts were significantly correlated to clinical variables pertaining to cancer stage and treatment efficacy. Albeit, I consider these findings to provide new insight into cancer-related fields, I still have some suggestions.

1, Most figures are highly professional; however, the authors should guide the readers to the meaning of the images appropriately; otherwise, it is likely to cause misunderstandings. Therefore, I suggest that the author consider revising these figure legends again.

2, In figure 1, The author presented a schematic of the project workflow, and described tRF counts were extracted from Mintbase v2.0 for 425 primary tumor samples and 36 adjacent normal samples. However, It would be much better if the authors could provide Graphical Abstract. for this research, I suggest that they can take a look at the recent paper in MDPI (PMID: 31331013, 34834441, 35625729)

3, In Fig 7A, the author presented the bar graph of the total number of tRFs associated with the 7 clinical variables that displayed significant correlations to tRF expression. The author may need to use other statistical analyses such as ANOVA to calculate the P-value for three or more groups of data, and please update the “Statistical Analysis” of the Method during further revision(PMID: 36555654, 32064155)

4, There are few typo issues for the authors to pay attention to; please also unify the writing of scientific terms. “Italic, capital”? For example, please unify “nonsmokers” or “non-smokers” in the manuscript.

5, Meanwhile, the font is too small for some of the current figures; please also revise these figures. 

Round 2

Reviewer 1 Report

The authors have addressed all of my comments and introduced the changes needed to clarify the points raised. I believe that the manuscript can be accepted for publication in the present form.